# Precision Recovery After Spinal Cord Injury: Integrating CRISPR Technologies, AI-Driven Therapeutics, Single-Cell Omics, and System Neuroregeneration

**DOI:** 10.3390/ijms26146966

**Published:** 2025-07-20

**Authors:** Răzvan-Adrian Covache-Busuioc, Corneliu Toader, Mugurel Petrinel Rădoi, Matei Șerban

**Affiliations:** 1Puls Med Association, 051885 Bucharest, Romania; razvancovache@innbn.com (R.-A.C.-B.); mateiserban@innbn.com (M.Ș.); 2Department of Neurosurgery “Carol Davila”, University of Medicine and Pharmacy, 050474 Bucharest, Romania; 3Department of Vascular Neurosurgery, National Institute of Neurology and Neurovascular Diseases, 077160 Bucharest, Romania

**Keywords:** spinal cord injury, neuroregeneration, molecular repair, precision medicine, gene editing, bioelectronics, neuroprotection, regenerative therapies, translational breakthroughs

## Abstract

Spinal cord injury (SCI) remains one of the toughest obstacles in neuroscience and regenerative medicine due to both severe functional loss and limited healing ability. This article aims to provide a key integrative, mechanism-focused review of the molecular landscape of SCI and the new disruptive therapy technologies that are now evolving in the SCI arena. Our goal is to unify a fundamental pathophysiology of neuroinflammation, ferroptosis, glial scarring, and oxidative stress with the translation of precision treatment approaches driven by artificial intelligence (AI), CRISPR-mediated gene editing, and regenerative bioengineering. Drawing upon advances in single-cell omics, systems biology, and smart biomaterials, we will discuss the potential for reprogramming the spinal cord at multiple levels, from transcriptional programming to biomechanical scaffolds, to change the course from an irreversible degeneration toward a directed regenerative pathway. We will place special emphasis on using AI to improve diagnostic/prognostic and inferred responses, gene and cell therapies enabled by genomic editing, and bioelectronics capable of rehabilitating functional connectivity. Although many of the technologies described below are still in development, they are becoming increasingly disruptive capabilities of what it may mean to recover from an SCI. Instead of prescribing a particular therapeutic fix, we provide a future-looking synthesis of interrelated biological, computational, and bioengineering approaches that conjointly chart a course toward adaptive, personalized neuroregeneration. Our intent is to inspire a paradigm shift to resolve paralysis through precision recovery and to be grounded in a spirit of humility, rigor, and an interdisciplinary approach.

## 1. Introduction

### 1.1. Overview of Spinal Cord Injury (SCI)

The spinal cord is often referred to as the “highway of life”, as it must effectively communicate between the brain and body in order for cognition to be translated into movement, sensation, and reflexes. The spinal cord is the architecture that translates cognition into physical function. It allows for coordinated movement and sensory experience [1]. Spinal cord injury (SCI) disrupts all of these critical communication pathways and produces a devastating level of impaired motor control, sensation, and autonomic function. SCI is much more than a medical diagnosis: it is a life-altering event that will not only change the injured person’s experience of daily living but will shape a new interpretation of their autonomy and coping mechanisms towards adversity in daily life and disability [2].

Worldwide, SCI has an extensive landscape, affecting between 250,000 and 500,000 people annually. In high-income nations, SCI is intact within a narrative usually characterized by speed and momentum: road traffic accidents and sports injuries commonly disproportionately affect young men [3], while in low-income countries, the narrative of occupational hazards and safety at work exists and can demonstrate risk factors associated with SCI occurring at work. With limited or no fall prevention protocols, any fall or accident is contextualized within the precariousness of survival. Yet, within this narrative of SCI, it is a changing story. This changing story acknowledges an aging population and how, with age, we know the fragility of life but, paradoxically, want the vitality of youth [4,5]. However, age also brings with it increased risk of more severe injury related to osteoporosis, spinal degeneration, and other illnesses that have severe roles in fall prevention. After decades of study, we can no longer ignore the cascading effects of “a fall”, especially for postmenopausal women with advanced bone loss. The injury from the fall, at times, is immediately catastrophic. The duality of trauma from youth and degeneration following injury with age is not culturally or socioeconomically limited. The stories will always share the plight of individuals impacted by SCI [6].

Nonetheless, numbers alone do not express the significant ripple effects of an SCI. For an individual living with an SCI, the injury is not the end of a line; it is entry into a journey that involves pain, rehabilitation, and adaptation [7]. Depression and anxiety are more than unwelcome fellow travelers; they can be active enemies on the road to recovery that create barriers to rehabilitation and reintegration. The burden on society is also significant. In the United States, the lifetime cost of care for an individual with a severe tetraplegia is USD 5.8 million, accounting for direct costs (acute care, assistive technologies, and rehabilitation) [8,9]. This represents the health care costs associated with SCI, but not the economics of loss of participation in the workforce or the hidden costs (e.g., lost income, respite, and job security) for caregivers, family members, and partners [10].

Thus far, we have relied on the classical SCI assessment with anatomical and functional variables: the anatomical level of the injury (i.e., cervical, thoracic, lumbar) and completeness of motor and sensory loss. These measures are anchor points; however, they do not account for the complexities of SCI in terms of the molecular and cellular levels. The advent of novel biomarkers, including neurofilament light chain (NfL) and glial fibrillary acidic protein (GFAP) in plasma and cerebrospinal fluid (CSF), is changing this narrative for SCI [11]. CSF GFAP levels seconds to hours following injury correlate with astrocyte reactivity and progression of lesion, while NfL in plasma represents a systemic biomarker in long-term neurodegeneration [12]. For example, elevated GFAP levels within 24 h of injury have consistently proven to correlate with poorer neurological outcomes and can serve as predictors of some chronic SCI trajectories, emphasizing their potential as biomarkers. Novel imaging modalities (such as diffusion tensor imaging (DTI)-based tractography) now display insights into the structure of axonal integrity, exposing underlying levels of white matter damage [13]. Combining that with functional MRI (fMRI), which identifies neural activity in association with functional recovery, and AI that interprets these datasets, we are beginning to have evidence to predict SCI trajectories and possible individualized therapeutic regimens utilizing AOIs [14,15]. This new approach reveals potential for first-generation precision diagnostics and therefore a break from the limiting view of SCI as a homogeneous pathophysiological entity and towards a model wherein it is understood as a set of molecular, structural, and functional perturbations.

While there are existing classification systems that assess SCI based on anatomical location and functional severity, these systems often neglect the considerable cellular and molecular heterogeneity that typifies patient-specific recovery outcomes. Advances in biomarker development and computational modeling are shifting how SCI severity is understood, in that they are leading to dynamic, real-time prognostic tools that go beyond a one-time neurological score.

Liquid biopsy technologies are identifying circulating molecular biomarkers of injury trajectories. Extracellular vesicles (EVs), including exosomes, can also carry injury-specific miRNAs and proteins, providing a less invasive way to monitor neuroinflammation, oxidative stress, and regenerative processes [16]. For example, elevated levels of exosomal miR-124 reflect microglial reprogramming, while miR-29 regulates extracellular matrix (ECM) remodeling and could affect glial scar formation. These molecular biomarkers can provide early indications of states of chronic inflammation and/or maladaptive repair, which would facilitate more targeted interventions [17].

Simultaneously, the field of metabolic profiling is identifying systemic metabolic alterations that appear to impact recovery trajectories. For instance, alternatives to glucose metabolism and/or mitochondrial lipid oxidation clearly increase secondary injuries, while changes in the kynurenine pathway indicate a role for neurotoxic metabolites in extended neurodegeneration. The identification of these metabolic shifts has opened new therapeutic windows and approaches, including the potential intervention of shifting metabolism to aid downstream cellular resilience [18].

To bring these molecular markers into clinical practice, machine learning models are being developed that survey biomarker data as they relate to neuroimaging signals to improve recovery predictions [19]. A comparison of patterns in DTI-based axonal integrity, fMRI connectivity, and serum biomarkers provides AI-based prognoses and individualized rehabilitation protocols, accounts for the timing of their intervention, and predicts long-term neurological function in this population more accurately than traditional assessments [20,21].

The coupling of liquid biopsy diagnostics, metabolomics, and AI-assisted interrogation of outcomes is changing SCI to a precision medicine model and, more broadly, care in general [22]. This first step in reframing and using precise diagnostic and prognostic tools will move us away from thinking about SCI as only an anatomical diagnosis and towards patient-specific molecular phenotyping.

This review is arranged along a translational continuum from pathophysiology to therapy. After describing the prevailing molecular mechanisms involved in the progression of SCI, including neuroinflammation, ferroptosis, oxidative stress, glial scarring, and metabolic reprogramming, we will not treat these mechanisms in isolation but as multi-faceted and dynamic contributors to impacted tissues, which provide both vulnerabilities and targets for therapies. Using this mechanistic context, we will then discuss next-generation interventions that could be used to modify these mechanisms: CRISPR gene editing, stem cell therapies, AI drug discovery, neuromodulation, and bioengineered scaffolds. In the final sections, we will bring together these molecular and technological elements to propose a precision medicine framework based on systems biology, single-cell omics, and neural digital twins for developing individualized and scalable approaches to spinal cord regeneration. With this organization, we hope to provide not only summaries of developments but also an integrated roadmap for future collaborative approaches to SCI recovery.

Ultimately, this work is necessary to bridge the gap between the acute injury and long-term therapeutics—this shift is further contextualized in the next section, where we will break down how we catalyze the underlying molecular dysfunction of neuroinflammation, cell death, and repair.

### 1.2. Importance of Molecular Understanding in SCI

SCI is more than just an insult to the body: it is a magnificent molecular dilemma in which destruction and repair take turns damaging and restoring normal function in the nervous system. The initial traumatic injury (the mechanical deformity of compression, contusion, or laceration) provides the harsh opening chord; the rest of the cascade of injury, cellular, and molecular events follow [23,24]. The challenge is the secondary injury phase—the storm that will unfold over hours and days, which ultimately determines the long-term outcome. Secondary injury is also a place of injury but potential, and the science seeks to reset the arc of pathology to recovery [25].

Neuroinflammation: The Paradox of Destruction and Repair

Neuroinflammation is the process underlying the secondary injury, which is one of the most complex and paradoxical of the four processes that unfold after CNS injury. Once the blood–spinal cord barrier (BSCB) is breached, the CNS immune-privileged space is transformed into a chaotic war zone [26]. Peripheral immune cells from the blood, including the neutrophils, macrophages, and T cells, invade the injury site, while microglia, the CNS’s resident immune cells, are primed, albeit not fully activated yet. Just as in skeletal muscle regeneration, the activation of microglia status changes through time:M1 Microglia: The cellular stormtrooper, a tissue-level synergy of chemical defense, predominantly from the TNF-α and IL-1β that are produced, with their main contribution being to the ongoing death of neurons, which includes bystander neuronal death [27];M2 Microglia: The cellular repairers, which contribute to recycling debris and initiate reparative changes by way of a suite of neurotrophic compounds, including IL-10 and TGF-β, to commence regeneration [28].

Microglial plasticity provides an exciting target to support neuroprotection and recovery. New technologies for nanocarrier-based delivery of CX3CR1 agonists are now beginning to demonstrate some potential for reprogramming microglia towards a reparative state, providing reduced lesion size and improved axonal regenerative behavior in preclinical models [29].

Oxidative Stress: The Silent Destroyer

Oxidative stress is the unseen antagonist of secondary injury, commencing a molecular barrage that may compromise cellular integrity. Reactive oxygen species (ROS) produced by defective mitochondria and NADPH oxidase are willful destroyers and catalyzers, causing fragmentation of DNA, lipid oxidation, and protein denaturation [30]. Antioxidants such as mitoquinone (MitoQ) and engineered superoxide dismutase (SOD) mimetics are possible molecular shields that help maintain mitochondrial integrity and eliminate ROS [31]. The Nrf2 pathway is the chief regulator for antioxidant defense and is gaining more interest as a clinical therapeutic target to drive cellular tolerance to oxidative stress [32,33].

Emerging Pathways: Ferroptosis and Systems Biology Insights

Further compounding classical pathways is the recently discovered ferroptosis, a cell death mechanism driven by lipid peroxidation in a manner similar to necrosis. Ferrostatins and iron chelators are useful inhibitors of this new cell death pathway [34]. In second injury research, computer modeling has yielded insights into unexpected interactions between necroptosis and inflammation, demonstrating how damage-related molecular pattern (DAMP) expression results in a feedback loop exhibiting destructive behavior. These findings are motivating the trialing of dual-pathway inhibitors as therapeutic interventions with the possibility of acting synergistically [35].

An approach to systems biology provides a requisite framework and methodology to investigate this molecular complexity. Researchers applying transcriptomics, metabolomics, and clinical imaging are using appropriate data to develop predictive models intended to relate to molecular signatures and recovery. These approaches should also facilitate pairing patient-specific molecular and clinical data to develop personalized therapeutic interventions leveraging AI capabilities, reducing the dissonance between cellular repair and functional recovery [36,37].

There is great excitement in the field of SCI research with unprecedented opportunity on the horizon. The integration of molecular biology, imaging techniques, computational modeling, and artificial intelligence is changing how we recognize and approach injury complexity. These new pathways and the use of integrated systems can bring much-needed illumination into a complex injury and ultimately influence best clinical practices to transform recovery and inspire hope.

Table 1 aims to serve as a conceptual introduction to the molecular pathophysiology presented in detail in Section 2. Instead of considering the mechanisms as distinct processes, we consider them overlapping injury and repair axes, whereby oxidative stress augments ferroptotic damage, while inflammation relays these states to modulate scar formation and vascular permeability. Section 2 categorizes these molecular cascades through the time continuum of SCI, from the changes associated with the biomechanical disruption and vascular component of primary injury to secondary injury responses such as metabolic failure, cell death, or glial remodeling. The structural design enables the reader to consider cause–effect relationships that inform windows of therapeutic opportunity in the future, as well as begin to build a foundation for consideration of the targeted therapeutic approaches to be discussed further in this review. By linking each therapeutic strategy with a specific type of failure in this cascade, we intend to move from providing a summary of the literature to facilitating synthesis—we do not expect to have the breadth of mechanisms separate from analytical discussion of their translational potential within one conceptual approach.

## 2. Molecular Pathophysiology of SCI

SCI triggers a series of molecular and cellular events that begin with the immediate mechanical damage and then progressively become an exceedingly complex secondary injury mechanism that results in further damage over time. Advances in molecular biology, nanotechnology, and other omics approaches have provided insight into these mechanisms and many novel mechanisms and therapeutic targets for SCI. This section will utilize the enhanced understanding of SCI to provide a deeper understanding of these dynamic processes and emerging concepts, avenues for new therapeutic options, and areas of translational potential.

### 2.1. Primary Injury: The Mechanical Trigger and Immediate Molecular Response

The physiologic primary injury phase occurs instantaneously. It is mechanical in nature, from forces including compression, contusion, or laceration—mechanical insults that destroy the structure and function of the spinal cord and set into motion a molecular mechanism of damage that perpetuates neural dysfunction for an extended time.

Biomechanical Complexity and Axonal Vulnerability

The spinal cord’s unique and anisotropic structural characteristics affect the way mechanical forces are distributed, with axons, in particular, in white matter tracts being particularly vulnerable to mechanical insult. Recent studies based on cryo-electron tomography suggest that the severing of axonal microtubules during SCI is coupled with nanoscale damages to the axonal transport machinery, particularly the motor proteins dynein and kinesin. As a consequence, mitochondrial transport and vesicular transport, which are necessary for survival, stop, resulting in more damage [58].

Recent results also show that new nanoscopic ECM molecules, such as heparan sulfate proteoglycans (HSPGs), also maintain structural integrity and biomechanical stability of the spinal cord. The HSPGs specifically also confer elasticity to the spinal cord, and their abnormal structures may increase vulnerability during a primary insult. Using bioactive peptides and glycan-modifying enzymes is being investigated to stabilize ECM components and minimize structural deterioration after a mechanical trauma [59].

Computational fluid dynamics models have afforded a view into the hydrodynamic forces during SCI and demonstrated how CSF shear stress in the subarachnoid space may exacerbate vascular injury [60]. These findings also open up next steps for potential targeted interventions to limit secondary vascular injury.

Therapeutics aimed at cytoskeletal integrity, for example, solvents targeting taxane derivatives, have expanded their possible use to include engineering microtubule-stabilizing nanoparticles as in vivo stabilizing agents. A distinguishing feature of nanoparticles amongst other stabilizing agents is their spatial specificity of delivery, or the delivery of a chemical stabilizing agent that contains spatial specificity, enhancing the possibility of utilizing chemical stabilizing agents in a way that maximizes potential therapeutic effect [61].

Vascular Dysfunction and Hypoxic Stress

The initial mechanical trauma is first thought of as primary injury to the spinal cord vascular system, where vascular complications will evolve as a continuum of injury to include de-formation of endothelial cells, rupturing capillaries, and breaking of the blood–spinal cord barrier (BSCB). The time course of evolving vascular complications can be further complicated by the immediate release of von Willebrand factor, due to the endothelial cell damage, and the resultant increases in clot formation, consequently severely limiting perfusion and increasing ischemia [62]. High-resolution imaging modalities, such as intravital confocal and two-photon microscopy, have recently defined a largely important role in the vascular permeability and immune cell invasion involved with disruption of the endothelial glycocalyx [63,64].

Therapeutic options aimed at vascular repair are evolving from previously targeting vascular stability with nanobody developed agents that are capable of stabilizing tight junction proteins from further degradation. In addition, glycocalyx restoration agents such as sulodexide are currently being investigated to restore the protective ability of endothelial cells and vascular longevity, indicating that they can be targeted for therapeutic interventions [65].

Hypoxia-inducible factor-1α (HIF-1α) is continually regulated in regions of hypoxia that limit cellular energy utilization by maintaining organisms in anaerobic metabolism using glycolysis. More recently, in several metabolomics studies, the accumulation of succinate is driving the inflammatory macrophages and subsequent production of mitochondrial ROS to further worsen injury pathophysiology. Future therapeutic advances would target succinate dehydrogenase in an effort to modify this inflammatory axis and protect energy metabolism at the same time [66].

Molecular Triggers: Calcium Dysregulation and DAMP Release

Increased calcium influx through stretch-activated ion channels (TRPV4, Piezo1, and ASIC1a) is a key immediate consequence of mechanical trauma. Calcium overload disrupts mitochondrial function, activates calpains, and instigates very rapid proteolytic degradation of cytoskeletal proteins, including spectrin and ankyrin [67].

Recent work has also established a link between calcium dysregulation and the phenotypes of endoplasmic reticulum (ER) stress, characterized by misfolded proteins contributing to eventual cellular dysfunction. Furthermore, necrotic cells release extracellular chromatin fragments, which activate the cGAS-STING pathway to drive the production of interferon and amplify neuroinflammation [68,69]. New therapies to shift the activation of STING by necrotic fragments to suppress this pro-inflammatory signaling cascade and other downstream responses are being investigated, including selective STING inhibitors [70].

Calcium chelators encapsulated in lipid nanoparticles are also currently being researched to capture excess calcium at the site of injury, which will reduce or prevent cytoskeletal collapse and will help better protect mitochondrial function [71].

Necroptosis and Autophagy: Divergent Cellular Responses

Necroptosis, a programmed necrotic pathway activated by RIPK1 and MLKL, contributes cumulative harm to the injury response since necroptosis causes release of intracellular membrane-bound immunogenic contents. To date, proteomic studies have identified phosphorylation sites on MLKL that can serve as markers of necroptosis activity, enabling fine-tuning of therapeutic intervention [72].

There is growing evidence that autophagy may promote or inhibit other pathways of cell death, including ferroptosis. While initially perceived as a protective process, autophagy can also intersect with deteriorated lipid degradation processes that leverage outcomes of lipid peroxidation. Therefore, one treatment strategy may be to combine autophagy enhancers with ferroptosis inhibitors like the selective inhibitor, liproxstatin-1 [73]. Current findings indicate that the mechanical force-dependent axonal vulnerability that exists in axons encapsulated by an ECM is influenced at the site level by microtubule lattice rigidity and that large-diameter axons rupture asymmetrically under tension. Cryo-electron tomography studies showed microtubule shearing at the site level, dynein–kinesin decoupling, mitochondrial mislocalizations, and loss of ATP, creating localized bioenergetic failure related to excitotoxic vulnerability [74].

Influencing the vulnerability are the structural fragility ascribed to the ECM’s rapid loss from the injury site, as the degradation of HSPGs and perineuronal nets exposes previously cryptic DAMP motifs that promote neuroinflammation. Computational modeling of CSF shear stress shortly after injury suggests that hydrodynamic turbulence contributes to ECM disorganization, cavitation, and barrier formation with subsequent astrocytic activation [75]. Novel glycan-engineered scaffolds are currently under investigation to mitigate biomechanical failure or redesign the ECM remodeling towards a pro-regenerative footprint [76].

Vascular imaging studies show that post-trauma, the endothelial glycocalyx collapse causes rapid pericyte dislodgement, microvascular rupture, and leukocyte eviction; that undirected lipidomic profiling of injured tissue reveals oxidized phospholipids (i.e., 4-HNE, MDA) as disruptive agents to the stabilizing zonula occludens-1 (ZO-1) against BSCB disruption [77]. Targeting these molecular disruptions with nanobody-based protective stabilizers and phospholipid sequestering nanoparticles is emerging as a promising strategy to limit acute ischemic toxicity [78].

At the ionic level, mechanosensitive ion channels (e.g., Piezo1, TRPV4) hyperactivate in milliseconds following injury and produce an immediate calpain-dependent cytoskeletal breakage with mitochondrial rupture, leading to opening of permeability transition pore (mPTP), cytosolic cytochrome efflux, and apoptosis. At the same time, released nuclear DNA fragments from necrotic neurons activate cGAS-STING signaling when integrated with astrocytic activation, creating an IFN-mediated neuro-inflammatory loop that perpetuates injury [79]. Recent advancements towards cGAS-STING inhibitors and blocking mechanosensitive channels have opened the door to possible early interventions to interrupt pathways unleashing the secondary injury molecular amplification [80].

The intersection of cytoskeletal breakage, ECM damage, vascular collapse, and Ca^2+^-dependent necro-inflammasome activation has initiated a cascade of injury that cannot be reversed and leads to secondary injury. The next section will discuss how oxidative stress, ferroptosis, and neuro-immune dysregulation contribute to the SCI disease process and the long-term consequences.

### 2.2. Secondary Injury: The Amplifying Cascade

Secondary injury path mechanisms occur over a matter of hours to weeks after the initial injury and cause secondary injury through the activation of various interlinked cascades, which expand the area of injury by creating an unfavorable microenvironment for repair and regeneration.

#### 2.2.1. Neuroinflammation: The Janus-Faced Immune Response

Neuroinflammation is a common characteristic of secondary injury and involves the interactions of the resident immune cells of the CNS with peripheral leukocytes that have infiltrated the area of injury. The timing of inflammatory responses has now begun to be characterized and has demonstrated that, after an injury, there are different stimulation response phases of macrophage cells that transition from pro-inflammatory M1 phenotypes to reparative M2 phenotypes [81].

Microglial Senescence and Plasticity

Recent studies have uncovered a geriatric population of senescent microglial cells that remain in a pro-inflammatory state, continually releasing cytokines like IL-6 and TNF-α, resulting in chronic inflammation behaving in ways resistant to existing anti-inflammatory strategies [82]. The development of therapeutic strategies focusing on microglial senescence, senolytic, and even inhibiting p16INK4a to reverse the senescent cells back to functioning microglia could decrease the inflammation [83].

Extracellular Vesicles in Inflammatory Propagation

Microglia and astrocytes both release extracellular vesicles (EVs) that carry pro-inflammatory content, including cytokines, as well as microRNAs. Exciting advancements in engineering extracellular vesicles have made it possible to deliver anti-inflammatory agents, such as miR-124 mimics, as a means to reprogram the immune response to be less inflammatory and more reparative [84].

#### 2.2.2. Oxidative Stress: The Molecular Wildfire

Oxidative stress remains a major component of secondary injury driven by mitochondrial injury, NADPH oxidase activity, and lipid peroxidation. Recent lipidomic studies have identified oxidized cardiolipin as a possible biomarker for oxidative stress in the mitochondria, which provides more options for diagnostics and treatment interventions [85,86].

Proteostasis and Oxidative Stress Interplay

Disruption of protein homeostasis creates additional oxidative damage through hateful assembly of misfolded proteins, where the emergence and development of proteasomal targeting enhancers can enhance the 20S proteasomal activity, introducing small-molecule activators to oxidatively protect against cellular dysfunction and pathological processes [87].

#### 2.2.3. Excitotoxicity and Programmed Cell Death

Astrogliosis or astrocytic dysfunction, such as calcium wave dysfunction, triggering excess glutamate release from excitatory or hyperactive astrocytic cells, perpetuating excitatory neuronal death, as target astrocyte-specific types of calcium channels, such as TRPA1, is another approach to decreasing excitotoxicity damage [88].

Programmed cell death mechanisms have begun to emerge, such as pyroptosis and ferroptosis, driving secondary modes of injury. The immunogenic character of ferroptosis mechanisms, being driven now by lipid peroxidation products, increasing recruitment of immune cells, is being targeted with therapy by combining gasdermin D inhibitors with inhibitors of ferroptosis pathways to facilitate modulating these inflammatory cell death pathways [89,90].

Emerging Concepts and Translational Perspectives

1.Adaptive Gene Editing: The future ability to adapt gene editing, such as CRISPR, to the future adaptive technological improvements of using base editing and prime editing will help adjust unnecessary inhibitory genes like CSPGs and reduce unwanted off-target effects [91];2.Systems Biology with SCI: The use of AI-based models allowing for changing omics data (including transcriptomic, metabolomics, and lipidomics) imaging as well as clinical outcomes through diagnosis and personalization of treatment, is pivotal in knowledge translation of SCI, better and more targeted subjects for future SCI recovery technologies [92];3.Therapies Embedded with Nanotechnology: The emerging field combining the application of lectins to understand different cell targeting and the multi-functional nanoparticles combining antioxidant and anti-inflammatory properties, as well as delivering growth factors, is poised to be the new precision/aligned therapeutics [93];4.Holistic approaches: Not just focusing on the body but on the entire body system’s rehabilitation, e.g., gut–brain axis, healing, and metabolic reprogramming, these are a new set of paradigms, although still undefined, utilizing a rehabilitation approach to recovery mechanisms [94].

The next table (Table 2) summarizes important new therapeutic approaches for recovering from SCI by detailing their approaches, process, and findings while identifying the available clinical evidence, therapeutic considerations, and limitations.

While Section 2 may have presented a number of molecular mechanisms implicated in SCI, including vascular instability, oxidative damage, and varieties of regulated cell death, it must be remembered that these processes are situated within a greater and dynamic biological framework. The interplay of glial cells, extracellular matrix components, and intracellular regulatory circuits can all dynamically interact and could determine the progression of injury or repair. The following section will explore these multilayered interactions, with the aim of clarifying how molecular signals may interact with the structural and cellular environment of an injured spinal cord. Rather than propose exclusive models, this perspective will attempt to integrate current evidence in a way that might be useful in developing more integrative approaches for future therapeutic research.

## 3. Multimolecular Interactions in SCI

The microenvironment following an SCI is a dynamic, multilayered array of cellular and molecular interactions contributing to the continuum of injury and repair. The possible interactions include glial responses, ECM, intracellular signaling, and epigenetic factors, which collectively determine recovery trajectory, create barriers to regeneration, and identify possible therapeutic targets. This section focuses on glial responses, inhibitory molecules, intracellular circuits, and epigenetic factors, synthesizing current understanding and identifying knowledge gaps to contribute to the development of an SCI therapy framework.

### 3.1. Glial Cell Responses: Commanders of Environmental Remodeling

Glial cells (astrocytes, microglia, and oligodendrocytes) are key players in SCI and are always holding a balancing act in the environment between repairing and inhibiting. Glial cells remodel the injury environment, thereby influencing the critical dimensions of tissue damage and repair following SCI. To mitigate glial response by augmenting their reparative potential and managing their inhibition, it is important to recognize the complexity of their biology.

#### 3.1.1. Astrocyte Reactivity and Glial Scar Dynamics

Astrocytes are generally the first responders to a SCI and are subject to a reactive transformation due to cytokines (e.g., interleukin-6 [IL-6], leukemia inhibitory factor [LIF]), and other factors that initiate the JAK/STAT3 signaling cascade, ultimately promoting proliferation and ECM synthesis [113]. Reactive astrocytes provide emergent underlying substrates, e.g., laminin and thrombospondin, which help stabilize the injury environment and provide short-duration ECM substrates permitting acute axon sprouting. While astrocyte reactivity is essential for acute response, excessive reactivity leads to the development of a glial scar. The glial scar is composed of a small-dense fibrous tissue with a high density of chondroitin sulfate proteoglycan (CSPG), which directly inhibits contralateral regenerative axon sprouting and blocks further sprouting [114].

New advances in single-cell RNA sequencing (scRNA-seq) methods allow for the identification of astrocytic phenotypes. Animal studies began to identify the neuroprotective feature of A2 astrocytes, which express Igf1, and the neurotoxic feature of A1 astrocytes, or complement factor C3. Extended activation of astrocytes increases the inhibiting properties of astrocytes, together with the resolution of the IL-33 response, contributing to rigidity in the scar [115]. Astrocytes also influence synaptic remodeling after impairment, impacting recovery, whether as a stabilizer of synapses through the secretion of thrombospondins, or a pruner of synapses that leads to a change in plasticity. Newer studies are moving to the astrocyte-neuron coupling of metabolism, including the transfer of functional mitochondria, through either EVs or gap junctions, to facilitate energy supply to the axon [116].

As a whole, there are presently astrocyte-specific therapies that are designed to either reprogram the astrocytic phenotypes or, conversely, manipulate the ability of astrocytes to interact with the ECM in subsequent recovery. Nanoparticle sustained-release administered chondroitinase ABC formulations are able to degrade CSPGs and disrupt the inhibitory signals from the astrocytes while keeping the astrogliosis protective [117]. Functionalized biomaterial scaffolds with neurotrophic factors [ex., brain-derived neurotrophic factor (BDNF)] have shown a positive connection and predictable integration with glial scars to support axonal regrowth. Engineered astrocyte-derived EVs with anti-inflammatory microRNAs like miR-21 and miR-124 may also be able to change the state of injury microenvironment and support tissue repair [118].

Knowledge Gap: Astrocyte-Vascular Crosstalk

The molecular signals that astrocytes and endothelial cells use to initiate vascular remodeling continue to be poorly understood. Understanding the influence of VEGF and angiopoietin-Tie2 signaling may lead to other target sites for therapeutic avenues to restore vascular integrity after SCI.

Astrocytes have also been shown to regulate extracellular acidity after SCI, and a loss of acidity equilibrium mediates axonal sprouting and inflammatory activation [119]. Astrocytic sodium–hydrogen exchangers (NHE1) or monocarboxylate transporters (MCTs) might modify proton gradients in a glial scar to shape neuronal death and ECM remodeling [120]. CSPG condensation induced by acidosis has been demonstrated to further increase the glial scar inhibitory response, creating additional limitations to axonal growth [121]. Novel therapeutic approaches have aimed to manipulate astrocyte regulation of pH by employing carbonic anhydrase inhibitors, creating a more permissive regime for neural regeneration [122].

A second area of interest is astrocyte mechanotransduction. Following an SCI, a reactive astrocyte changes its mechanical properties and produces cytoskeletal stiffening by actomyosin contraction, which adversely affects their neurotrophic support of neurons and their gap junction connectivity, reducing the exchange of metabolites and ions with the neurons [123]. Some studies have indicated the use of blebbistatin derivatives to inhibit actomyosin contractility as a possible mechanism to return astrocytes to a more plastic state and restore their functional integration with neural networks [124].

#### 3.1.2. Microglial Dynamics and Immunometabolism

Microglial responses to injury after SCI are multifaceted and, in part, influenced by the previously discussed M1/M2 polarization model. There is increasing appreciation in the field that multiple transcriptionally distinct populations of microglia are activated in their different functional states after spinal cord injury as a result of transcriptional plasticity. Single-cell transcriptomics have sequenced injury-associated microglia (IAMs), lipid droplet-accumulating microglia (LDAMs), and neuroprotective homeostatic populations of microglia with distinct metabolic and epigenetic programs responsible for driving respective phenotypes [125]. IAMs are characterized by the expression of Hif1α, Trem2, and ApoE and remain in a continuously neurotoxic and inflammatory state while exacerbating tissue damage. They identified a reparative microglia subpopulation that had an upregulation of Arg1, Chil3, and PPAR signaling, which was involved in removing cellular debris outside of the spinal cord and in the remodeling of synapses [126].

Microglial activation leads to a temporal and spatial variable activation pattern after SCI as opposed to an on/off response. Chronic inflammation is associated with lipid metabolic dysfunction in microglia, which aggregate oxidized lipids in order to mediate neuroinflammation [127]. As a means to “reprogram” microglia to return to their homeostatic state, there are available pharmacological and preclinical options such as PPARγ agonists and Trem2-modulating treatments. Furthermore, pharmacological specific inhibition of the Hif1α-mediated repopulation of pro-inflammatory microglia for the treatment of chronic SCI may be a possibility to potentially slow the progression of secondary injury [128]. This opens the door for selective microglial modulations that will be specific improvements of the subtype of microglia that were defined from the specific single-cell level phenotype versus broader M1 or M2 phenotyping categorization.

Aside from these classical roles for inflammation, microglia have the ability to actively maintain spinal cord homeostasis in the role of a mechano-sensory cell. There is more evidence evolving that changes in mechanical properties of viscoelastic stiffness within the tissue following SCI can have downstream effects on the activation state of microglia. Stiffening ECM will elicit a chronic neurotoxic microglial phenotype [129]. The mechanosensitive change of the microglia is transduced through integrin-FAK signaling pathways, which may accelerate the sensing of the stiffness gradients and enable the transduced microglia to activate, from a non-responsive stance, a maladaptive over-expressing chemical signal that involves proinflammatory gene expression. In contrast, certain mechanical loads may positively affect the microglia’s clinical transition from neurotoxic to a reparative phenotype on soft new biomaterial scaffolds that may mimic the elastic properties of the spinal cord through loads that use the stiffness of the scaffold as a loading parameter [130,131].

The newly revealed mechanism of chronic microglial activation includes lipid scavenging dysfunction. Disallowed lipid mechanisms lead to accumulating oxidized phospholipids as a result of those lipidic processes, paired with NF-κB persistence and cytokine release [132]. Nonetheless, several options are being explored to “reprogram” lipid metabolism of the microglia through some pharmacological options, including liver X receptor (LXR) agonist treatments, which promote mitochondrial transport of lipids to suppress neurotoxic inflammation and restore homeostasis [133].

#### 3.1.3. Oligodendrocyte Death and Challenges in Remyelination

Oligodendrocytes provide myelinated and metabolic support to axonal fibers and are likely to be impacted by secondary injury or choices such as ferroptosis, oxidative stress, and excitotoxicity. Ferroptosis occurs with lipid peroxidation and is now emerging as the most prominent cause for oligodendrocyte death and subsequent remyelination, as seen with markers on ACSL4 and GPX4 in increased SCI tissue [134,135]. Oligodendrocyte death causes demyelination and ultimately loss of saltatory conduction while increasing the metabolic burden on the axons, ultimately causing loss of function.

Oligodendrocyte progenitor cells (OPCs) are recruited through chemokine gradients such as CXCL12 and mobilize to the injury site. When OPCs arrive at the spinal cord, there are limitations to differentiation into oligodendrocytes due to inhibition of the ECM molecules and impaired mitochondrial function. Lipidomic approaches have established depletion of the essential phospholipids required by oligodendrocytes to synthesize myelin, which limits remyelination. As demonstrated in Figure 1, the demyelination/remyelination processes include maturation of oligodendrocytes and macrophages/microglia.

New and potential treatment paradigms aim to either update OPC function or overcome barriers imposed by the ECM. For example, the alternative use of oxidative stress-resistant mitochondria as a means for mitochondrial transplantation has shown promise in returning OPC energy metabolism so they can differentiate and provide support for remyelination processes during neurotrauma. In addition, small molecules targeting Wnt-signaling pathways (e.g., inhibitors of GSK-3β) have demonstrated effectiveness in promoting OPC differentiation and remyelination in traumatic brain injury and SCI models. In addition to targeting lipid synthesis, inhibition of lipolysis may allow oligodendrocytes to re-establish the metabolic balance necessary for efficient myelin production [136].

Knowledge Gap: Axon-OPC Interactions

The signaling pathways that regulate OPC differentiation in relation to the functional activity of axons and whether these pathways change based on levels of neuronal activity are poorly understood. Future studies can examine the activity-dependent modulation of OPC maturation and myelination through glutamate signaling.

With the aforementioned emphasis on oligodendrocyte depletion or ECM inhibition after SCI as disadvantages for remyelination, recent evidence suggests that disturbances in lipid droplet metabolism distinctly hamper myelin repair [137,138]. Oligodendrocytes utilize lipid turnover for producing (including cholesterol) myelin sheaths, but SCIs induce inflammation, which alters lipid recycling pathways, thus accumulating reduced peroxidized lipids that are toxic to OPCs [139,140]. Further, there is evidence that pharmacologically activating sterol regulatory element-binding proteins (SREBPs) can establish myelin lipid synthesis, a prior identified molecular target for precision remyelination [141].

Another covert target in SCI remyelination failure is misregulation of ion channels. In the case of OPCs and oligodendrocytes, Kv1.3 and Kir4.1 potassium channels are expressed and play key roles in myelin sheath integrity and oligodendrocyte metabolic stability [142]. Since SCI causes oxidative stress in the spinal cord, the channels in the oligodendrocytes convert from membrane binding to become internalized, which disrupts ionic homeostasis [143]. Current work from our lab and others is directed towards optogenetically reactivating oligodendrocyte channel-specific potassium channels as a strategy designed to restore bioelectrical communication of oligodendrocytes with axons [144].

### 3.2. Inhibitory Molecules: Molecular Barriers to Axonal Growth

Inhibitory molecules like myelin-associated inhibitors (MAIs) and CSPGs are the predominant inhibitory molecules in the environment of an injured spinal cord, and they throw several molecular hurdles in the way of axonal regeneration through a mechanism by modulating neuronal receptors and intracellular signaling accompanying the cellular responses [145].

#### 3.2.1. Myelin-Associated Inhibitors

Myelin-associated inhibitors (MAIs) include molecules like Nogo-A, MAG, and OMgp that bind to neuronal receptors like NgR1 and PirB, causing the activation of the RhoA/ROCK pathway, which ultimately inhibits actin polymerization and disassembles the cytoskeleton. Ephrin-B3 is also an MAI that engages in greater inhibition when bound to Eph receptors [146,147].

Neutralizing antibodies against Nogo-A have progressed to clinical trials and appear to have promise for promoting axonal sprouting and recovery of function. Combinations of dual-targeting antibodies against Nogo-A and Ephrin-B3 have even greater regenerative effects due to their synergistic effects. Finally, decoy receptors that integrate soluble fragments of NgR1 delivered through engineered exosomes could be useful for inhibiting MAI inhibition [148].

#### 3.2.2. Chondroitin Sulfate Proteoglycans

CSPGs, which are secreted mostly from reactive astrocytes, can inhibit regenerative responses when bound to neuronal receptors like PTPσ and LAR to inhibit the cytoskeleton. In the subacute and chronic phases of SCI, super-resolution imaging showed dynamic deposition of CSPGs, which formed a biochemical barrier for axonal regeneration [101,149].

Targeting CSPG by CRISPR-based gene editing of the enzymes responsible for the biosynthesis of CSPGs has been one approach for decreasing inhibition from scars after SCI. More recently, ECM scaffolds are being developed that incorporate both CSPG degradation and growth factor delivery to promote axons down permissive pathways [150].

Recent work points to the possibility that CSPG signaling may be attenuated through modulation of glycosylation enzymes instead of complete CSPG degradation. The enzymatic removal of CSPGs appears to be unnecessary because just changing chondroitin 6-sulfation to a different glycosylation pattern may neutralize axonal growth inhibition and maintain structural integrity of the ECM [151]. The novel CRISPR-based glycoengineering strategies for sulfotransferase (such as CHST3 and CHST15) provide a promising avenue for researchers to modulate ECM turnover without disturbing paths to healing. This route has possibilities because it may alleviate excessive glial scarring and the uncontrolled rebuilding of CSPGs, but without eliminating or limiting beneficial components of the ECM [152].

An additional factor that has emerged is the relative viscoelasticity of the matrix and the additional role it plays in CSPG-mediated inhibition. After injury, the ECM solidifies with fibrotic proteins (like fibulin-5 and transglutaminase-2) cross-linking [153]. Simply softening the ECM with bioengineered hydrogel-based inhibitors has reduced inhibitory properties through the ability to restore mechanical plasticity to the ECM. This mechanical approach to modulating ECM stiffness offers a novel way of modulating the inflammatory response to healing an SCI (even as an adjunct to bio-chemical targeting of CSPGs) [154].

### 3.3. Intracellular Signaling Pathways: Translating Inhibition into Regeneration

As mannered processes, when regenerative signaling reaches cells, it moves through multiple intracellular signaling processes, which are critical bottlenecks for axonal regeneration. The RhoA/ROCK pathway, which is a key signaling process for cytoskeletal inhibition, has been largely successful for ROCK-pathway inhibitors like fasudil to help axonal growth in rodents in preclinical studies [155]. While there are numerous new technologies in development, we can firmly state that optogenetics can yield basic improvements using spatiotemporal activation of intracellular signaling events, PI3K/Akt, ultimately resulting in famous improvements in mitochondrial performance (AKT role with apoptosis, cell growth/cytoskeletal changes, and mitochondrial function), cytoskeletal changes, and neuronal survival [156].

An emerging regeneration axis in SCI regeneration is the relationship between Hippo signaling and its downstream effectors, YAP/TAZ, via governing axonal cytoskeletal organization, mitochondrial organization, and growth cone dynamics. Reports have shown that YAP/TAZ signaling enhances microtubule cytoskeletal stability as a result of MAP1B phosphorylation and directed localized ATP production at growth cones, thereby facilitating neurite extension [157]. Unfortunately, the SCI environment activates Hippo signaling to abnormally high levels, sequestering YAP/TAZ and disrupting regeneration. Proof of concept has supported developing gene therapy strategies that would reactivate YAP/TAZ in preclinical models, improving axonal regeneration and functional improvement [158].

### 3.4. Future Perspectives

Emerging technologies promise new paradigms for SCI research and therapies. Synthetic gene circuits are being programmed to release neurotrophic factors or anti-inflammatory cytokines in response to injury-relevant stimuli. Spinal cord organoids combined with microfluidics allow researchers to simulate SCI and test interventions [159]. AI-based models can plan for combination strategies, using omics and imaging in order to optimize recovery. Activating plasticity using other methods, such as electrical stimulation and wearable rehabilitation devices, represents a potential for promoting functional recovery [160].

## 4. Potential Recovery Mechanisms and Therapies

The task of overcoming SCI has advanced into a multidisciplinary effort, with molecular biology, bioengineering, and regenerative medicine united for the same goal: repair and repair. What used to be deemed an impossibility is now being reframed as a complicated, resolvable problem, with every discovery allowing a glimpse into new pathways to recovery. By targeting the immune response and dynamics, axon regeneration, neuroprotection, and genetic modulation, modern therapies will change SCI research and patient treatment around the world.

### 4.1. Modulating the Immune Response: Turning Adversaries into Allies

The immune response to SCI is complicated and paradoxical, and it is crucial in clearing debris and mediating repair; however, in circumstances of chronic inflammation, it can damage the injured tissue. Therapeutic strategies used to focus on minimizing the immune response. Today’s thought has shifted to enhancing the beneficial aspects of the immune response whilst dampening the detrimental parts. This has led to using a differential approach to post-injury modulation [161].

In response to SCI, the BSCB opens, resulting in a robust pro-inflammatory response normally mediated by cytokines including TNF-α, IL-1β, and IL-6, leading to oxidative stress and neuronal apoptosis. While broad anti-inflammatory approaches have been taken in the past, including steroids such as methylprednisolone, the immunosuppressive effects of widespread therapies have encouraged the use of refined approaches [161]. Current therapies involve monoclonal antibodies targeting TNF-α (e.g., infliximab) and IL-1β (e.g., canakinumab), which add a conscientious inflammatory response suppression while generating alternative immune system pathways for repair [162].

Recent advances in immune profiling, like recent advances in single-cell RNA sequencing (scRNA-seq) and spatial transcriptomics, have taken us further forward in discovering what immune response types are activated following SCI while also revealing further diversity [163,164]. These tools allow for the identification of reparative immune subpopulations, which include regulatory T cells (Tregs) and anti-inflammatory macrophages, and some of these populations can be selectively expanded and/or reprogrammed for therapeutic ends. Tregs expanded with IL-2/anti-IL-2 complexes can resolve inflammation and promote tissue repair, and macrophages are also being polarized from a pro-inflammatory M1 phenotype to a reparative M2 state using cytokines (e.g., IL-10 and TGF-β) or by administering engineered exosomes containing IGF-1 [165].

Recently, physical rehabilitation and electrical stimulation have emerged as adjuncts to immune modulation to support the establishment of immune tolerance, mediated by Treg stability and macrophage polarization (shown to build synergistically towards repair). These synergistic pathways are good examples of highly translational molecular and physical interventions. In addition, there are biomaterials, such as injectable hydrogels that contain anti-inflammatory compounds or cytokines that can buffer immune activity, which can work with advancing neuroprotective biomaterials to achieve neural repair, while also lending themselves to some sort of immune modulation [166]. These developments will represent the next generation of therapies that will couple unheard-of medicine with the precision of molecular therapies and physical and structural assistance [167].

### 4.2. Overcoming Axonal Growth Inhibition: A Path Through Barriers

Discovering ways to promote axonal regeneration following spinal cord injury rehabilitation is the holy grail of recovery, but the presence of molecular inhibitors in the microenvironment of the injury will continue to impede recovery. MAIs and CSPGs have formed a biochemical barrier to recovery, forcing acceptance of alternate pathways to repair [168].

MAIs such as Nogo-A, MAG, and OMgp act on neuronal receptors NgR1 and PirB, which activate the RhoA/ROCK pathway, which we know negatively impact growth cones and disturb cytoskeletal structure. In preclinical studies, neutralizing antibodies (e.g., ATI355), which are very promising because they promote both axonal sprouting and dissociation of the development of function, have emerged. With advances in structural biology, small-molecule inhibitors are being developed to block MAI-receptor interactions, and CRISPR-based gene-editing technologies are being used to silence NgR1 to create what we hope are permanent solutions. Molecular chondroitin sulfate proteoglycans (CSPGs), derived from reactive astrocytes, represent the main building blocks of glial scars because they form a dense extracellular matrix (ECM) made up of solids and semi-solids that can block axonal extension in a chemical way. Chondroitinase ABC is the gold-standard enzyme for degrading CSPGs and has been loaded into nanocarriers and hydrogels to optimize its stability and spatial localization and reduce postponed dosing. Or CSPG neutralizing peptides can block receptor actions while still allowing ECM disruption [149,169,170].

The time-dependent dynamics of inhibition show that CSPGs peak during the sub-acute phase, but MAIs are prolonged, and they can persist into chronic phases, showing that different stages need different interventions. To provide some context to the above and previous findings, activity-based interventions associated with inhibitory-targeting therapies, such as electrical stimulation, for example, and electrical stimulation + chondroitinase ABC, led to increased axonal extension and greater functional recovery by initiating growth-promoting circuitry. There are now emerging therapies that appear to target mechanosensitive channels for these reasons, like Piezo1, to motivate axonal regeneration. What is interesting about these therapies is that mechanical loads are used to mimic cytoskeletal dynamics + cellular events, thus gaining the advantage for axonal regeneration [171,172].

### 4.3. Neuroprotection: Preserving What Remains

In recovery from SCI, protection is as valuable, if not more valuable, than recovery of surviving neural tissue from the continuum of medications of secondary injury. These events together will also help nullify and exacerbate the condition of the injured individual, increase the injury zone, and decrease recovery [173]. Therefore, neuroprotection becomes the foundation of therapeutic interventions.

Mitochondria-targeted antioxidants such as MitoQ and SS-31 are helping to change how we treat oxidative stress through improved mitochondrial status + preventing energy deficits [174,175]. Combined with NMDA antagonists (i.e., memantine), we can reduce oxidative stress and excitotoxicity through a dual-protection strategy [176]. Catalytic antioxidants, such as cerium oxide nanoparticles, mimic endogenous enzyme systems to generate a robust source of ROS scavenging to protect both neurons and glia [177,178].

Ferroptosis, a novel form of cell death related to lipid peroxidation, was a significant contributor to oligodendrocyte death in SCI, and ferroptosis inhibitors (e.g., liproxstatin-1) may limit white matter degeneration and facilitate improved functional outcomes [179,180]. At the same time, targeted ER stress therapies (e.g., salubrinal and ISRIB) may limit protein misfolding and subsequent cell insult during the chronic phase of SCI [181,182].

Injectables are also capitalizing on this format; one noteworthy example is self-healing hydrogels. Hydrogels utilize biomechanics to adapt to the injury; they release antioxidants, caspase inhibitors, or neurotrophic factors in response to injury-specific stimuli [183]. Their persistent release ability and ability to respond dynamically to internal and external stimuli features could be protective of the neural tissue and create conditions that sustain regenerative therapies [184].

### 4.4. Stem Cell Therapies: Rebuilding from Within

Stem cell therapies epitomize the regenerative medicine field due to their ability to replace lost tissue, alter the microenvironment of injury, and promote axonal regeneration. Mesenchymal stem cells (MSCs) are preferred, particularly due to their immunomodulatory and neurotrophic capacity, as they secrete neuroprotective factors such as BDNF, VEGF, and IGF-1, promoting neuronal survival as well as angiogenesis. MSC-derived EVs are a safer, indirect way of transplantation, as EVs are trafficked bioactive molecules to deliver to the host; in addition, they have a far lower likelihood of immune reaction [185,186,187].

Induced pluripotent stem cells (iPSCs), also derived from patient-specific cells, are changing personalized medicine. iPSCs can then differentiate into neurons, oligodendrocytes, and astrocytes to replace lost or reduced cells when repair of the tissue is required [188,189]. Advancements in scaffold-based delivery systems are improving cell survival and integration with cells derived from iPSCs, and CRISPR-based genetic editing has the potential to solve enhancement concerns related to tumorigenicity. Neural organoids from iPSCs are now being tested as therapeutic constructs and models to test regenerative interventions [190].

As the clinical use of stem cell engineering for SCI develops, ensuring genetic and oncological safety will be paramount. Considerable oncogenic risk could arise due to the presence of any residual undifferentiated iPSCs, specifically due to teratoma development during engraftment, especially if purification does not adequately remove pluripotent-associated contaminants [191]. While CRISPR-based edits are said to be locus-specific, there is an opportunity for cryptic off-targeting effects, including indels, translocations, and p53 pathway activation, which may have previously gone unnoticed but pose susceptibility towards malignant transformation [192]. Many of the mutations arising from CRISPR editing will evade detailed detection with standard PCR or Sanger sequencing, which is why there is a suggestion for a more sensitive form of detection, such as GUIDE-seq, high-fidelity Cas variants, and whole-genome single-cell analysis [193]. There are authorities that are now encouraging quality control pipelines, including karyotyping, vector integration assays, telomerase activity, and long-term xenograft assays for tumorigenicity. The only way we can truly approach a clinically relevant application of stem cell-derived neuroregeneration is to link and define multi-layered biosafety checkpoints [194].

The role of bioengineered scaffolds is becoming increasingly sophisticated, using scaffolds that are electrically conductive and smart biomaterials, which bridge stem cells and the spinal cord injurymicroenvironment. These scaffolds only act as a new mechanism for cell delivery, providing structural support and facilitating cell differentiation and human axonal growth, better demonstrating a next-generation device for functional recovery [195].

### 4.5. Gene Therapy: Precision Healing

Gene therapy is introducing a method of ‘-action’ on spinal cord injury that focuses on the subsequent activation of the physical trauma treatment mechanisms of the injury by examining the mechanisms of the injury treatment at the molecular level, working on the premise of a gene therapy treatment paradigm ‘a molecular change will alter the outcome of the injury.’ CRISPR/Cas9 tools are enabling selective gene editing, dictating transcription alterations on NgR1 and biosynthetic enzymes in a manner that will change the injury microenvironment from non-supportive to a supportive environment, promoting axonal regeneration. Base editing and prime editing are the latest technologies in the lineage of CRISPR/Cas9 technologies—these allow but do not require molecular breaks in the gold standard of CRISPR, for corrections to select bases and base pairs, while having lower off-target effects [196].

RNA-based therapies that are mounted as a biologic, targeting a diverse set of pro-inflammatory miRNAs (miR-NAs), targeting and/or activating miRNA ‘antagomirs’ (e.g., miR21) and/or and bioengineered miRNA regenerative mimics (e.g., miR124), are being delivered in lipid nanoparticle formulations that rapidly clear the delivery system (e.g., surfactant) while providing local and sustained local performance. Epigenome editing with CRISPR-dCas9 systems can dictate molecular genome changes, with X epigenome editing systems evolving rapidly in terms of how the methods employed can activate regenerative pathways or silence inhibitory genes.

The complementary introduction of gene therapy to activity-dependent approaches such as optogenetically controlled stimulation is a unique new direction to spinal cord injury recovery. Both methods not only repair damaged pathways but also rewire functionality in the neural circuits (e.g., distinct action potentials to synaptic connectivity changes) to promote functional recovery in spinal cord injury.

### 4.6. Artificial Intelligence in SCI Therapy: Precision, Prediction, and Personalization

In exciting ways, AI is revolutionizing SCI therapy by merging multi-omics data, scientific drug design, and adaptive neuromodulation. Machine learning algorithms that are trained on incredibly high-dimensional biological datasets can reveal multi-target drug interactions, optimize compounds for neuroprotection, and even allow for a patient-specific anticipated response [197]. Graph neural networks (GNNs) can map injury-specific molecular networks and identify potential novel therapeutic targets, beyond current pathway analysis routinely utilized in the field, paving the way for novel therapies altogether, while reinforcement learning algorithms can optimize drug structures for bioavailability and blood–spinal cord barrier permeability [198].

While AI-based models that predict multi-target drug interactions may be more accurate and complex, they have not been validated in biological contexts. For example, graph neural networks and reinforcement learning-based frameworks are promising for ranking neuroprotective compounds based on favorable blood–spinal cord barrier permeability, though these predictions have not been validated in biological assays, high-throughput screening, or in vitro efficacy, yet they have not been verified in a wet-lab for verification or validation, which is a very important step in the clinical translation [199]. The next logical step will be to embed phenotypic screening platforms, automated drug testing in neural organoids, and single-cell readouts to limit the number of tested compounds to those highly predicted to interact with therapeutically relevant end-points with actual activity. A translational pipeline using in silico inference for virtual screening, followed by real-time biological validation, will be an important step in realizing the true therapeutic potential of SCI with AI [200].

AI will also soon have an impact on biomaterials engineering and cell therapy. For example, generative models may now facilitate the design of bioactive scaffolds by combining possible porosity structures, degradation rates, and release profiles for growth factors or connectors. Furthermore, AI-guided single-cell transcriptomics can reliably predict optimal stem cell differentiation trajectories, theoretically reducing the chance of post-transplantation failure [201]. With respect to computational rehabilitation, there is the concept of a neural digital twin—a simulated AI-derived version of the patient’s nervous system that is a dynamic replica of the patient made up of their embedded neuroimaging, electrophysiology, and molecular data to simulate how an injury could have progressed. These digital models would not just be static simulations: they would continuously update in real time so that rehabilitation protocols can be constantly optimized as well as to explore combinatorial therapy options, for example, an adaptive neuromodulation parameter [202].

The use of neuromodulation is also evolving. Knowledge on the advantages of using AI-guided closed-loop brain–machine interfaces (BMIs), as well as spinal cord stimulation (SCS), continues to grow. Now, advanced deep learning algorithms can decode neural signals with exceptional accuracy, allowing the possibility of controlling neuroprosthesis through AI-guided reinforcement learning [203]. In addition, AI-guided phase-locked neuromodulation shows promise, using externally applied electrical stimulation synchronized to endogenous cortical oscillations to magnify plasticity and ultimately return voluntary motor control. Furthermore, these systems utilize the real-time neural activity to adaptively adjust stimulation levels to maximize the therapeutic benefits while minimizing maladaptive plasticity [204].

While the opportunities of AI in this field are exciting, there are challenges that must be tackled with respect to data standardization, interpretability of various standard measures, and no algorithm is perfect and imparts biases. Explainable AI (XAI) frameworks will be crucial to providing more clinical visibility into solutions to clinical decision making, as well as exploring federated learning models that potentially improve data integration in a multi-center data model without compromising patient safety and privacy concerns. This also highlights the importance of ethics in AI in regard to wording and decision making in rehabilitation protocols. As the potential for a more automated rehabilitative-driven approach to decision-making emerges in clinical practice, we want to ensure it does not replace clinician oversight and that the final responsibility remains with the clinician [205].

Combining computational neuroscience, regenerative medicine, and real-time neuromodulation is influencing translational change in our SCI therapy practices from one focused on empirical decisions to mediate injury recovery, i.e., bedtime rehabilitation, to an adaptive, precision modeling-driven practice. This change will happen as machine intelligence continues to hone and develop predictions, moving our understanding of SCI therapy from a reactive approach to a continuing, evolving patient-specific model [206].

## 5. Challenges and Future Directions

This is an exciting time of innovation in SCI research, spurred on by advances in molecular biology, regenerative medicine, bioengineering, and computational sciences. Issues of recovery potential and potential life-saving interventions abound with emerging possibilities. However, the challenge lies in whether we can actually transform the knowledge from preclinical science to clinical practice, which ultimately can influence the lives of patients. Translational research presents the first barrier to overcome, and of course, translating preclinical discoveries to address spinal injuries is complicated by the systemic and neurological nature of SCI, requiring individual patient solutions. In this section, we discuss translational barriers, the ethical implications of our work, combination or synergistic therapies, and emerging technologies that have the potential to forever change the way we study or treat SCI.

### 5.1. Translational Barriers: From Models to Clinical Realities

Transitioning from preclinical success to clinical efficacy remains one of the significant barriers in SCI research. The use of animal models of spinal cord injuries often does not fully recreate the intricacies or pathophysiology of human spinal injuries, indicating that alternate platforms are necessary to enhance translational relevance [207].

The use of organoid systems in combination with microfluidic chip systems, called organoid-on-a-chip technology, is a powerful way to model SCI. These devices are paramount as they recreate human-relevant environments that utilize vascular flow, nutrient gradients, and inflammatory responses [208]. These systems have already uncovered new therapeutic targets by simulating the interactions of vascular and neural injury in human-like conditions.

Case Study: Identifying neuroprotective compounds with organoid-on-a-chip systems. Researchers used an organoid-on-a-chip model to identify a compound designed to target the Nrf2 antioxidant pathway. When tested, it produced a ~50% reduction in ROS-induced neural injury compared with traditional antioxidants and greatly improved neuronal survival. The results from the Phase 1 clinical trials demonstrate the innovations in translation via the excellent platforms [209].

iPSC models from patients are allowing the opportunity for personalized medicine, generating spinal organoids that allow every individual to replicate their i-jury and understand drug responses.

A case study: iPSC models for therapeutics designed with precision. Investigation, using patient-specific spinal organoids, derived two distinct patient organoids (minor vs. major inflammation) and demonstrated that patient organoids derived from (minor) inflammation patients demonstrated increased sensitivity to IL-6 cytokine inhibitors, as was predicted, compared to their control. The evidence generated from supporting personalized therapeutics helped to further pre-clinical therapies in their follow-up study and was attributed to a 35% increase in the number of responders to the drug [210].

To speed up the change of these ideas and platforms being developed into clinical applications, they will need to overcome the existing challenges to manufacture them. The new bioreactors and unique cell culture systems will provide the scalability of each product produced and ensure reproducibility after each round. Unfortunately, the heterogeneity of patients limits the design of clinical trials. A solution may be to provide a combination of adaptive trial methods and clinical design methods and strategies, using biomarkers such as GFAP and NfL, that can enable trial paradigms of credible adaptive possibilities.

Transition: All together, these examples point to the fact that while there is a significant gap in translation, the new platforms and uniquely personalized ways are creating a useful bridge/headway from bench research to clinically applied practice.

### 5.2. Novel Non-Coding RNA Targets

Non-coding RNAs (ncRNAs) interact to address critical roles throughout SCI pathophysiology, including inflammation, apoptosis, and axonal regeneration. Their unique stabilization, versatility, and multifunctionality make them an attractive player in responsive therapeutic delivery systems.

A case study: CircHIPK3 is a mechanistic candidate to drive axonal regeneration. CircHIPK3 was used for engineered exosomes in a rat SCI model, and potential axonal regeneration was increased by +45%, and a specific motor function performance improved by > +70% of the induced baseline. CircHIPK3 demonstrated both improved neurotrophic signaling and modestly reduced inflammation [211].

The possibilities for therapeutic delivery systems are proceeding at stunning speed. RNA nanostructures, capable of delivering multiple ncRNA molecules simultaneously, provide an opportunity for combined effects by targeting multiple pathways.

Example: RNA nanostructures for microglial modulation. Those RNA nanostructures that delivered mimics of miR-124 produced a 40% reduction in microglial activation, maintained synaptic integrity, and improved functional recovery scores in a chronic SCI model. This example illustrates that ncRNA approaches can be more specific and effective than traditional anti-inflammatory approaches [212].

Broader Insight: New advances in research of ncRNA are advancing in a way that you will likely eventually be able to not only combine therapeutic ncRNAs with gene-targeting tools such as CRISPR that downregulate target gene expression and inflammatory pathways but also manipulate both targets simultaneously for recovery success.

### 5.3. Advanced Stem Cell Engineering

Stem cell therapies are still considered the gold standard in regenerative approaches to address SCI. Unfortunately, there is little to invent or improve upon in stem cell therapies. The advances in genetic engineering, synthetic biology, and bioprinting should only provide further enhancements to stem cell therapy for some time to come.

Example: CRISPR-enhanced neural stem cells. In a rat SCI model, the CRISPR-/Cas9-edited NSCs (which overexpressed GDNF) demonstrated enhanced survival, reduced apoptosis, and localization to the injured site. The authors also demonstrated that GDNF-edited NSCs improved post-surgical motor functional scores by 30% compared to non-edited control NSCs, which is evidence that gene-edited NSC therapies may be highly effective for SCI [213].

Innovative delivery systems are also playing a crucial role in advancing stem cell therapies.

Case Study: Bioprinting of spinal constructs in large animal models. In a porcine model, integration of iPSCs in biospinal constructs resulted in the first major advancement toward personalized bioengineered approaches for SCI recovery, as shown by a 75% restoration of electrophysiological activity and some remyelination [214].

Future Direction: The combination of omics-guiding design with AI-enabled predictive modeling is likely to further optimize stem cell engineering, enabling patient-specific therapies tailored to individual genetic make-up and injury particulars.

### 5.4. Bioelectronic Interfaces

Bioelectronic technologies can be viewed as a new form of neuron rehabilitation therapy that combines neural stimulation with molecular therapies.

Case Study: Combining spinal cord stimulation with RNA therapy. A clinical trial combining SCS with RNA therapy targeting Nogo-A showed 25% greater motor recovery than SCS alone. The synergistic effects of bioelectronics and molecular interventions underscore the potential of hybrid approaches in SCI treatment [215].

Future Insight: Developments of hybrid platforms that incorporate bioelectronic stimulation with a real-time chemical release system targeting neurotrophic factor delivery are likely to be a significant departure from the personalized treatment repertoire.

### 5.5. Neuroprotective Innovations

It is possible to “preserve” neural tissue from secondary injuries and to promote “successful recovery” in the context of spinal cord injury. Ferroptosis, or lipid peroxidation, is one trajectory to target therapeutically.

General Case Study. Liproxstatin-1 decreased ferroptosis-related neuronal cell death by 60% in chronic rodent models of SCI. In comparison to animals not treated with Liproxstatin, there was clear documentation of improved locomotion and reduced lesion volumes, thus establishing ferroptosis as a valid therapeutic target [216].

Systemic Integration: The gut–brain axis is gaining traction as an important modulator of SCI recovery.

Case Study: A clinical trial involving interventions with probiotics targeted specifically to the microbiome reduced systemic inflammation by 30% in SCI, while improved bowel and bladder function in persons with SCI indicates the broad systemic health impacts on the recovery of neural tissues from spinal cord injury [217].

### 5.6. Ethical and Regulatory Considerations

The implications of rapid innovation highlight the need for ethical considerations and regulations.

General Case Study. A clinical trial of CRISPR-edited stem cells in spinal cord injury highlighted the importance and need for long-term monitoring plans to assess the effects of off-target. The initiative introduced international guidelines to monitor genetic safety as a credible approach to regulate “safe” innovation to ensure it is equitable and scalable [218].

### 5.7. Future Technologies

The convergence of progressive technologies to innovate a new model of intervention for spinal cord injury is ongoing.

General Case Study. In a study of quantum discovery of drugs for spinal cord injury, computation of quantum simulations revealed the potential for several drug combinations targeting oxidative stress and neuroinflammation. The significant result for speeding up timelines for drug discovery was a 40% reduction of delays for the drug combinations. The innovative work illustrates the transformative potential of quantum-driven directions for drug discovery [219].

Case Study: Wearable biosensors and a neuromodulatory system within a patient rehabilitation program reduce all rehabilitation timelines by 20% for persons with spinal cord injury. Wearable devices enabled real-time feedback, allowing clinicians with data to change rehabilitation protocols in real-time while progressing recovery outcomes [220].

## 6. Conclusions and Future Directions

SCI research is at a critical time in history, and with the prospect of integrating molecular biology, regenerative medicine, bioengineering, and computational research to change the boundaries of recovery, SCI research has evolved from a focus on understanding the complex pathophysiology of SCI to the testing of investigational therapies that aim to extend the potential of biological repair and functional restitution. Therefore, there is a clear change in the future that integrates new technology and interdisciplinary partnerships, and the prospect of the journey from paralysis to recovery feels less like a lofty dream and more like an axiom that has now materialized as a promising horizon.

### 6.1. Summary of Key Molecular Mechanisms

Unraveling the Complexity of Injury

The pathophysiology of SCI is a continuum of molecular and cellular events at multiple scales, as there are non-linear injury and repair pathways. While we see a primary injury, regardless of how it was delivered—compression, contusion, or laceration—it begins an infinite sequence of simultaneous and in-sequence concurrent processes such as calcium dysregulation, oxidative stress, and damage-associated molecular patterns (DAMPs). Each of these potential mechanisms of secondary injury can also begin processes of neuroinflammation, programmed cell death, and excitotoxicity that lead to an expansion of the injury footprint, but simultaneously can also yield some underlying mechanisms for therapeutic intervention.

Neuroinflammation as a Double-Edged Sword: The nasty property of neuroinflammation, as both destructive and restorative, has emerged as a key target for intervention, which involves re-skilling neuroinflammation through re-programming of resting glial cells—microglia—to the anti-inflammatory M2 phenotype; harnessing the destructive inflammation; and tapping into it as a facilitator for axonal regrowth for tissue restoration and repair.

Controlling or Targeting Oxidative Stress: ROs targets the tissue damage in the nervous system because it undermines the promise of the therapeutic hinge point. Recently, more mitochondrial armorers and ferroptosis inhibitors have emerged for a neurological—as well as clinical—closure target area to try to mitigate harmful outcomes. E.g., to date, in pre-clinical studies, the ROS-induced oxidative damage has been reduced by 60% as an end-point outcome.

Utilizing Gene Editing: CRISPR-based methods aimed at inhibitory molecules (such as CSPGs and the protein Nogo-A) are not only eliminating barriers to axonal regeneration but also providing supportive microenvironments for the repair process. New information, such as the role of circular RNAs (circRNAs), in synaptic plasticity and axonal regeneration, has expanded the possibility of therapeutic molecular targets. Some circRNAs, like circHIPK3, are driving RNA-based therapies, while systems biology and the use of omics technologies are allowing researchers to identify intervening pathways in the context of each injury category.

### 6.2. The Path Ahead

Interdisciplinary Collaboration for Complex Challenges

Recovery from SCI is an interdependent process; therefore, a system-based approach is needed to fully understand its molecular, cellular, and systemic interconnected mechanisms. Through interdisciplinary collaboration, what was once the advance of individual disciplines can now be combined to counter the complexities of SCI recovery by bringing together the experts: bioengineers, computational scientists, clinicians, and neuroscientists.

A Unified Framework: By creating complex recovery systems through the combination of bioelectronic interfaces, stem cell therapies, and molecular diagnostics, it offers recovery strategies specific to the individual patient. For example, recovery profiles involving omics-guided profiling and computational modeling allow for personalized treatments that conform to the unique biological fingerprints of the injury, which is characteristic of that individual’s SCI.

Accelerating Clinical Translation

Despite the advances in preclinical research, moving into the clinical application phase has continued to be difficult. Early-phase clinical trials are using advanced technology to help transition from this supposed yawning chasm.

Biomarker-Driven Trials: The utilization of biomarkers [i.e., GFAP and NfL] allows for more reliable patient stratification and maximizes the potential of conducting the clinical trial in cohorts that are most likely to demonstrate the success of the therapy being investigated. Furthermore, the implementation of adaptive trial designs supports speeding up the validation of new therapies by decreasing time and cost to develop the new therapies. Rehabilitation Integration: Studies that combine both molecular therapies and advanced rehabilitation technologies like robotic exoskeletons or wearable biosensors are raising the new bar of recovery by incorporating functional gains along with molecular repair.

Enhancing Quality of Life

Recovery is also more than just restoring motor function. New interventions that target their systemic health, such as probiotics to modulate the gut–brain axis, are decreasing both systemic inflammation and autonomic dysfunction, thus improving both physical and emotional health. New neuroprosthetics, along with cognitive rehabilitation programs, aim to substantially restore independence, sense of agency, and purpose for an individual.

### 6.3. Emerging Trends and Opportunities

Synthetic biology is enhancing the development of programmable cells and biomaterials based on different injury environments. These engineered systems can detect changes occurring in inflammation, oxidative stress, and nutrient availability, which trigger the release of therapeutic agents in real time. The emergence of these sorts of technologies, collectively referred to as “smart technologies”, will undoubtedly create a new era of intelligent, self-regulating treatments.

Bioelectronic hybrids are enabling the seamless integration of molecular and electrical cues to enhance neural repair. These interfaces are increasingly being paired with computational models that predict optimal stimulation patterns, personalizing recovery protocols.

Neural Digital Twins: These virtual replicas of a patient’s spinal system are being used to test therapies and optimize rehabilitation strategies, significantly reducing trial-and-error approaches in clinical settings. Quantum simulations are revolutionizing SCI research by modeling the complex molecular interactions that govern injury and repair. Recent studies have shown that quantum-driven drug discovery platforms can identify synergistic combinations of neuroprotective agents, reducing preclinical timelines by up to 40%. Big strides in combining omics-guided profiling with AI-driven analytics are offering unprecedented potential for personalized therapies. Using machine learning models, we will conduct meaningful deconvolution of neurotrophic factors, cytokines, and pharmacological agents to optimize combinations for each patient’s unique bio-signature and injury profile.

### 6.4. Unresolved Challenges

While there have been impressive advances in SCI recovery, challenges still exist in the field.

Translational Gaps: Closing the gap from preclinical to clinical application will require real improvement in experimental models and scalable biomaterials.

Global Equity/Accessibility: The clinical translatability of AI-based neurorehabilitation platforms, neural digital twins, and closed-loop bioelectronic implants is in serious danger of excluding the health systems in low-resource settings because of capital costs, the requirement for simple-to-use data infrastructure, and specialized training. Emerging models will emphasize equitable access by prioritizing frugal innovation, modular scalability, and decentralized integration. For example, edge-AI architectures—in which computation is performed on low-power chips (locally) rather than relying on power-hungry cloud servers—provide real-time processing of biosignals (e.g., EMG, EEG, HRV) to avoid threats of unreliable broadband or reliance on centralized GPUs. AI-guided rehabilitation may ultimately be viable in mobile and remote clinics; similarly, open-source neuromodulation systems (e.g., OpenBCI) along with non-invasive spinal stimulation platforms (e.g., focused ultrasound or transcutaneous) will create scalable interfaces without the surgical risk and cost. Moreover, the use of 3D-printed, biodegradable scaffolds and injectable biomaterials composed of alginate, chitosan, or silk fibroin provides feasible and inexpensive alternatives to the costly lab-based processes of hydrogel fabrication/bioactivity has been shown in SCI models. Integration with point-of-care manufacturing (e.g., portable bioprinters or lyophilized hydrogel precursors) could localize manufacturing to communities. Federated learning frameworks are now in pilot projects for data storage and training decentralized AI models across multiple sites—this permits models to learn from multicenter SCI datasets without the need to transfer raw patient data, thus preserving privacy while simultaneously building global generalizable algorithms. Transitioning from technology to sustainable equity necessitates clinical translation. This includes policy alignment by the WHO on medical device certifications of the LMIC members; medical devices with regional (tertiary) manufacturing hubs offering reliable, affordable hardware; and a mechanism for south–south collaborative networks to share collaborators for regulatory, engineering, or clinical contributions. Ultimately, neuroregenerative medicine must apply a “design-for-disparity” ethic, where innovation is not only appraised for the sake of frontier innovation but for its exercising ingenuity across increasingly problematic boundaries of infrastructure, income, and geography.

Monitoring Long-Term Outcomes: For gene-edited patients or patients with bioelectronic implants, it will take real effort to build proper frameworks to measure long-term safety and efficacy for the lifetime of the patient.

Oppressive ethical considerations must be managed as the headway is being made in SCI interventions. Reducing inequality to access new technologies, protecting patient privacy as AI-driven therapy infancy, and developing a global framework to guide the rights of gene-editing applications are all crucial to securing the rights of patients and public trust.

### 6.5. A Vision for the Future

In the next 10 years, SCI recovery promises a transformation into a converging integrative domain of the molecular diagnostic, regenerative therapy, and bioelectronic intervention within integrated care frameworks. Future therapies will adapt care protocols to their respective recovery trajectories by bringing together real-time assessments of their recovery and personalized interventions and adapting rehabilitation frameworks to support the maximum possibility of recovery and the potential of who they can be.

We must create scalable and affordable strategies to make sure SCI advances can be reached and included for every patient without socio-economic and geographic limitations. When academic institutions, private industry, and global health organizations collaborate, we can make SCI advancements democratized and accessible to all patients, wherever they may be. Recovery can no longer be measured only in clinical terms, but in independence, dignity, and resilience. If we are guided to the horizon by patients with SCI that we focus on, we will look ahead and re-imagine the possibilities for recovery and ensure that millions have hope.

Recovering from SCI is not an abstraction anymore: it is a space for action, innovation, and disruption that is full of opportunities. In the next decade, as technological advances grow, our understanding will explode, and the interest will grow exponentially in the exciting and emerging promise of recovering completely. It should become a reality, not something that is fleeting. The intersection of science, technology, and human ingenuity will allow us to create opportunities to dream about a future that is not limited by the problems that defined our past. This transformation is more than an innovation. It is a recognition that collaboration and engagement across disciplines and patient-centered concepts while addressing the disparities of the global health equities with advanced therapies to patients will all be important, and the change moving forward for all patients who will benefit from transformative discoveries will be actionable realities, with the promise that every patient who is recovering will have access to the tools they require to ultimately recover. Together, we will create a future that is defined by recovery and hope for millions, using science, resilience, and bold actions.

## Figures and Tables

**Figure 1 ijms-26-06966-f001:**
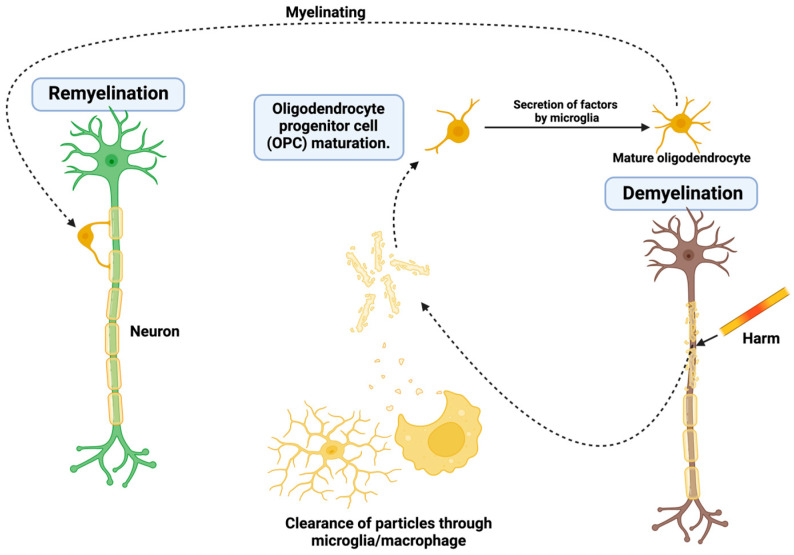
Demyelination and remyelination processes in SCI. The figure depicts the interplay between demyelination, characterized by oligodendrocyte loss and ECM barriers, and remyelination, mediated by OPC maturation and microglial-/macrophage-driven clearance of debris. Emerging therapies aim to enhance remyelination by targeting metabolic and signaling pathways. However, temporal dynamics are critical: acute degradation of inhibitory ECM components like CSPGs may destabilize lesion architecture, while prolonged depletion risks compromising structural ECM integrity.

**Table 1 ijms-26-06966-t001:** This table highlights pivotal molecular mechanisms in SCI recovery, focusing on pathways like ferroptosis, neuroinflammation, oxidative stress, and excitotoxicity, which drive injury progression and regeneration challenges. These mechanisms often do not act in isolation—for example, lipid peroxidation in ferroptosis amplifies pro-inflammatory M1 microglial responses, while persistent neuroinflammation further sensitizes neural cells to ferroptotic death via ACSL4 induction and GPX4 suppression. Therapeutically, ferroptosis inhibitors like liproxstatin-1 preserve white matter and support remyelination, while reprogramming neuroinflammation toward a reparative M2 phenotype using agents such as IL-10 holds synergistic promise.

Reference	Molecular Pathway	Key Findings	Mechanism of Action	Therapeutic Target/Agent	Implications for SCI Recovery	Limitations
[38,39,40]	Ferroptosis	Lipid peroxidation drives oligodendrocyte death; ferroptosis inhibitors reduce cell death by 50%	ROS-mediated lipid damage; ACSL4 overexpression in SCI	Ferroptosis inhibitors (e.g., liproxstatin-1)	Preserves white matter; supports remyelination	Off-target effects and systemic toxicity concerns
[41,42,43]	Neuroinflammation	Reprogramming microglia from M1 to M2 reduces inflammation by 40%	CX3CR1 agonists shift microglial phenotype	Nanocarriers delivering IL-10; CX3CR1 agonists	Harnesses immune response for repair; promotes axonal regrowth	Microglial plasticity timing is critical for therapy
[44,45,46,47]	Nrf2 pathway activation	Reduced ROS-induced damage; enhanced mitochondrial resilience	Activation of Nrf2-dependent antioxidant pathways	Antioxidants (MitoQ, SS-31)	Protects against oxidative stress; improves recovery timelines	Bioavailability of agents in CNS remains a challenge
[48,49,50]	Excitotoxicity	Inhibiting glutamate-mediated excitotoxicity preserves neuronal viability	Blockade of NMDA receptors	NMDA receptor antagonists (memantine)	Reduces neuronal apoptosis; neuroprotective	Potential cognitive side effects
[51,52,53]	Apoptosis pathways	Reduced neuronal apoptosis by 30% with caspase inhibitors	Inhibition of caspase-3 and -9	Caspase inhibitors	Prevents secondary injury-induced neuronal death	Target specificity and systemic side effects
[54,55,56,57]	Endoplasmic reticulum (ER) stress	Reduces protein misfolding; promotes neuronal survival	Modulation of UPR signaling	Salubrinal, ISRIB	Enhances neuronal survival	Long-term safety of ER-targeted agents is unknown

**Table 2 ijms-26-06966-t002:** Key combinatorial therapeutic strategies in SCI, selected for their mechanistic complementarity across neuroregeneration pathways. This table aims to synthesize recent advances in gene editing, nanotechnology, stem cell transplantation, immunomodulation, and ECM re-engineering. Each pairing is rooted in convergent pathology—for instance, combining ferroptosis inhibitors with IL-10 delivery targets both oxidative cell death and pro-inflammatory microglial activation. The table outlines therapeutic approach, methodology, supporting evidence, biological rationale, and limitations, offering a systems-level perspective for multimodal SCI intervention.

Therapeutic Approach	Mechanism/Methodology	Key Findings	Model/Evidence	Therapeutic Implications	Limitations	Reference
CRISPR-Cas9 editing targeting Nogo-A	Gene editing via CRISPR-Cas9 combined with activity-based therapy	40% axonal sprouting; improved motor recovery	Preclinical: Rodent (SCI + treadmill)	Restores neural connectivity; potential for chronic SCI	Off-target effects and long-term gene stability	[95,96,97]
Nanoparticle delivery of ChABC	Local nanocarrier-mediated delivery of chondroitinase ABC (ChABC)	CSPGs; glial scar; 60% axonal growth	Preclinical: Chronic SCI rat model	Combines precision delivery with matrix remodeling	Nanoparticle stability in systemic delivery	[98,99,100]
MSC transplantation with biomaterial scaffolds	3D-printed scaffolds seeded with MSCs + rehab protocols	70% motor function in large animals	Preclinical: Porcine SCI model	Enhances integration and axonal regrowth	Immune rejection risk and scalability issues	[101,102,103]
Mitochondrial transplantation	Injection of isolated healthy mitochondria into lesion	oxidative stress; preserved white matter	Preclinical: Rodent model	Novel acute-phase neuroprotective approach	Targeting and distribution challenges	[104,105,106]
Antioxidant-loaded nanoparticles	Nanoparticles encapsulating SOD mimetics or MitoQ	ROS levels; axonal survival	Preclinical: Murine SCI	Shields tissue from oxidative damage	Requires precise targeting	[107,108,109]
Combined gene therapy and exercise	Gene therapy suppressing CSPGs + treadmill training	45% functional recovery	Preclinical: Rodent model	Shows synergistic potential of genetic + physical therapy	High cost and logistics	[110,111,112]

## Data Availability

The data presented in this study are available on request from the corresponding author.

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
