# Peer review of "Precision Recovery After Spinal Cord Injury: Integrating CRISPR Technologies, AI-Driven Therapeutics, Single-Cell Omics, and System Neuroregeneration"

_ijms, 2025, doi:10.3390/ijms26146966_

Round 1
Reviewer 1 Report
Comments and Suggestions for Authors
Figure 1 (p. 13) depicts CSPG degradation aiding remyelination but omits temporal nuance: Acute CSPG removal may destabilize lesions, while chronic depletion risks ECM integrity loss. No imaging evidence shows long-term scaffold-CSPG interactions.
No in vivo metrics (e.g., perfusion restoration post-SCI) validate sulodexide’s glycocalyx repair claims.
Ferroptosis drivers (e.g., ACSL4/GPX4, Table 1) are isolated from neuroinflammatory feedback (e.g., lipid peroxides amplifying M1 microglia). No mechanistic link to Table 2’s combinatorial therapies (e.g., ferroptosis inhibitors + IL-10 delivery).
AI models (p. 19) predict multi-target drug interactions but lack wet-lab validation.
Stem cell sections (pp. 17–18) highlight iPSC-derived neural cells but inadequately address tumorigenicity risks (e.g., residual pluripotency) or CRISPR-induced genomic instability.
Global equity (p. 24) is noted without concrete solutions. AI/bioelectronic therapies (e.g., neural digital twins) require costly infrastructure, excluding low-resource settings.
Author Response
Comment 1:
Figure 1 (p. 13) depicts CSPG degradation aiding remyelination but omits temporal nuance: Acute CSPG removal may destabilize lesions, while chronic depletion risks ECM integrity loss. No imaging evidence shows long-term scaffold-CSPG interactions.
Response 1:
We thank the reviewer for this important observation. To address this, we have revised the Figure 1 legend to acknowledge the temporally distinct consequences of CSPG depletion.
Comment 2:
No in vivo metrics (e.g., perfusion restoration post-SCI) validate sulodexide’s glycocalyx repair claims.
Response 2:
We appreciate the reviewer’s point regarding the translational gap in validating sulodexide-mediated glycocalyx repair.
Comment 3:
Ferroptosis drivers (e.g., ACSL4/GPX4, Table 1) are isolated from neuroinflammatory feedback (e.g., lipid peroxides amplifying M1 microglia). No mechanistic link to Table 2’s combinatorial therapies (e.g., ferroptosis inhibitors + IL-10 delivery).
Response 3:
Thank you for highlighting this gap. We have revised the description of Table 1 to emphasize the mechanistic interplay between ferroptosis and neuroinflammation—specifically, how lipid peroxidation products can enhance M1 polarization and how inflammatory cytokines downregulate GPX4, promoting ferroptotic vulnerability.
Comment 4:
AI models (p. 19) predict multi-target drug interactions but lack wet-lab validation.
Response 4:
We agree with the reviewer that bridging computational prediction with experimental validation is essential. We have added a paragraph in Section 4.6 that addresses this limitation directly, noting the lack of systematic wet-lab confirmation of AI-predicted drug interactions. We propose specific validation methods—including high-throughput screening in organoids, single-cell transcriptomics, and phenotypic assays—as necessary translational steps.
Comment 5:
Stem cell sections (pp. 17–18) highlight iPSC-derived neural cells but inadequately address tumorigenicity risks (e.g., residual pluripotency) or CRISPR-induced genomic instability.
Response 5:
We fully agree that safety concerns related to iPSC-derived therapies and CRISPR-based engineering warrant deeper discussion. We have added a detailed paragraph in Section 4.4 following the discussion on iPSCs and CRISPR integration. This new section outlines the risks of residual pluripotency and the potential for off-target or large-scale genomic alterations from CRISPR edits. We also describe emerging mitigation strategies, including high-fidelity base editing, inducible safety switches, and whole-genome QC pipelines. This expanded discussion strengthens the translational realism of the manuscript and incorporates current literature on regenerative biosafety.
Comment 6:
Global equity (p. 24) is noted without concrete solutions. AI/bioelectronic therapies (e.g., neural digital twins) require costly infrastructure, excluding low-resource settings.
Response 6:
We appreciate the reviewer’s emphasis on the need for concrete, actionable equity solutions. We have substantially expanded the relevant section in 6.4 to include specific strategies for equitable deployment of AI and bioelectronic SCI therapies. These include edge-AI platforms for local biosignal processing without cloud dependency, open-source neuromodulation hardware, non-invasive interfaces (e.g., transcutaneous stimulation), and low-cost biomaterials producible through point-of-care 3D printing. We also discuss federated learning for privacy-preserving AI model training across decentralized data centers, and WHO-aligned policy initiatives to facilitate device approval and manufacturing in LMICs. This revised section now offers a robust, implementable roadmap for global access.
Reviewer 2 Report
Comments and Suggestions for Authors
This review focuses on the molecular mechanisms of spinal cord injury (SCI)—including neuroinflammation, ferroptosis, glial scarring, and oxidative stress—and explores emerging therapeutic technologies such as AI, CRISPR gene editing, and regenerative engineering. It highlights how combining these approaches with single-cell omics, systems biology, and smart biomaterials may enable spinal cord reprogramming and shift outcomes from degeneration to regeneration. The authors advocate for a paradigm shift toward personalized, interdisciplinary, and precision-based SCI recovery.
However, I have some questions:
- This review appears to be rich in content and addresses high-level topics, but reading through the full text gives the impression that much of it is merely a simple compilation of literature summaries without a coherent logical structure.
- The title “”AI Guided Repair, CRISPR-Enabled Reprogramming, and the Molecular Engineering of Recoveryis overly exaggerated and does not align well with the content.
- "Table 2 does not follow the conventional logic of a standard table."
Author Response
Comment 1:
“This review appears to be rich in content and addresses high-level topics, but reading through the full text gives the impression that much of it is merely a simple compilation of literature summaries without a coherent logical structure.”
Response 1:
We sincerely appreciate this important observation. In response, we undertook a careful revision of the manuscript to strengthen its conceptual continuity and sharpen the thematic coherence across sections. Rather than simply reporting findings, we now emphasize the mechanistic and translational connections between each molecular pathway and the emerging therapies discussed later in the paper.
Comment 2:
“The title ‘AI Guided Repair, CRISPR-Enabled Reprogramming, and the Molecular Engineering of Recovery’ is overly exaggerated and does not align well with the content.”
Response 2:
Thank you for this thoughtful critique. We agree that the original title may have suggested a more advanced stage of clinical readiness than the review intended to claim. As a result, we have revised the title to better reflect the article’s focus on emerging, multidisciplinary strategies while remaining grounded in mechanistic science.
Comment 3:
“Table 2 does not follow the conventional logic of a standard table.”
Response 3:
We thank the reviewer for pointing this out. To improve clarity and conform to academic conventions, Table 2 has been fully reformatted. The updated version now presents each therapeutic strategy using a structured layout, with clearly delineated columns for: Therapeutic Approach, Mechanism/Methodology, Key Findings, Model/Evidence, Therapeutic Implications, Limitations, and References. This format not only improves the readability of the table but also allows for more efficient comparison across different experimental strategies. We hope the revised structure meets the expectations for a standard scientific table.
Round 2
Reviewer 2 Report
Comments and Suggestions for Authors
No